# Biosynthesis of Vanillin by Rational Design of Enoyl-CoA Hydratase/Lyase

**DOI:** 10.3390/ijms241713631

**Published:** 2023-09-04

**Authors:** Qi Ye, Weizhuo Xu, Yanan He, Hao Li, Fan Zhao, Jinghai Zhang, Yongbo Song

**Affiliations:** 1School of Life Sciences and Biopharmaceuticals, Shenyang Pharmaceutical University, 103 Wenhua Road, Shenhe District, Shenyang 110016, China; yeokqi@163.com (Q.Y.); heyanan987@outlook.com (Y.H.); 15694142939@163.com (H.L.); zhaofan@syphu.edu.cn (F.Z.); 2School of Functional Food and Wine, Shenyang Pharmaceutical University, 103 Wenhua Road, Shenhe District, Shenyang 110016, China; weizhuo.xu@syphu.edu.cn

**Keywords:** vanillin, biosynthesis, rational design, enoyl-CoA hydratase/lyase, molecular dynamics

## Abstract

Vanillin holds significant importance as a flavoring agent in various industries, including food, pharmaceuticals, and cosmetics. The CoA-dependent pathway for the biosynthesis of vanillin from ferulic acid involved feruloyl-CoA synthase (Fcs) and enoyl-CoA hydratase/lyase (Ech). In this research, the Fcs and Ech were derived from *Streptomyces* sp. strain V-1. The sequence conservation and structural features of Ech were analyzed by computational techniques including sequence alignment and molecular dynamics simulation. After detailed study for the major binding modes and key amino acid residues between Ech and substrates, a series of mutations (F74W, A130G, A130G/T132S, R147Q, Q255R, ΔT90, ΔTGPEIL, ΔN1-11, ΔC260-287) were obtained by rational design. Finally, the yield of vanillin produced by these mutants was verified by whole-cell catalysis. The results indicated that three mutants, F74W, Q147R, and ΔN1-11, showed higher yields than wild-type Ech. Molecular dynamics simulations and residue energy decomposition identified the basic residues K37, R38, K561, and R564 as the key residues affecting the free energy of binding between Ech and feruloyl-coenzyme A (FCA). The large changes in electrostatic interacting and polar solvating energies caused by the mutations may lead to decreased enzyme activity. This study provides important theoretical guidance as well as experimental data for the biosynthetic pathway of vanillin.

## 1. Introduction

Vanillin (4-hydroxy-3-methoxybenzaldehyde, CAS 121-33-5) is one of the most important flavoring agents in the world. Presently, the industrial vanillin product could be manufactured in three routes. Firstly, the classical natural vanillin can be extracted from natural vanilla beans [1]. Secondly, various chemical synthesis processes could be applied to generate vanillin from diversified sources, such as lignin and cresols [2]. For the complement/alternation of the unstable source and high costs of the natural vanilla pods, a third route which had been explored for years, and researchers had attempted multiple pathways to generate vanillin by biosynthesis with different raw materials, such as eugenols, isoeugenol and ferulic acids [3,4,5].

As early as 1977, Tadasa had reported the vanillin biotransformation from eugenols by *Corynebacterium* sp. [6]. Later on, more studies had disclosed the same biotransformation results with different microbes [7,8,9,10]. More extensive works indicated that isoeugenol could also be used as a substrate to generate vanillin [8,11,12], and enzymatic processes were also pointed out [13,14]. When ferulic acid was used as starting materials, multiple microbes had been tested and provided valuable genetic and enzymatic results for the vanillin biosynthesis pathways. The pathways could be classified into CoA-independent [15] and CoA-dependent (Figure 1).

In the CoA-independent pathway, the phenolic acid decarboxylase (Pad) and aromatic dioxygenase (Ado) could transform the ferulic acid into vanillin without coenzyme A [15]. This pathway also converts many other renewable lignin-related aromatics into valuable chemicals and pharmaceutical intermediates, thus enabling value-added utilization of lignin. In the CoA-dependent pathway, a gene encoding an enoyl-CoA hydratase/lyase (Ech) enzyme for the hydration and non-oxidative cleavage of feruloyl-CoA to vanillin had been isolated and characterized in 1998 [16]. The gene encoding feruloyl-CoA synthase (Fcs) was studied in *Pseudomonas* sp. HR199 (DSM7063) [17]. By using these two above genes, vanillin was been generated in engineered *E. coli* [18], *Rhodococcus* strains [10], *Pseudomonas* strains [19,20], *Pediococcus* strains [21], *Bacillus* strains [22,23] and *Streptomecyes* strains [24,25]. However, few reports have been disclosed for these enzymes rational design to improve their activities, resulting in the catalytic activity of these enzymes gradually becoming a bottleneck in the vanillin production pathway. Recently, with the development of computer-based technologies in the biocatalysis field, enzyme rational design by bioinformatics tools can effectively improve the catalytic activity of the enzyme.

In the study, due to the lack of reliable structural data for Fcs, only the Ech from *Streptomyces* sp. strain V-1 was selected as the enzyme to be modified. A various of bioinformatics techniques, including multiple sequence alignment, sequence conservation analysis, homology modeling, molecular docking and molecular dynamics simulation to clarify the molecular mechanism of the enzyme-substrate complex, and nine mutants (F74W, A130G, A130G/T132S, R147Q, Q255R, ΔT90, ΔTGPEIL, ΔN1-11, ΔC260-287) were ultimately generated by rational design to enhance production of vanillin. Whole-cell catalysis was performed for wild-type and all mutants using the pET-Duet-1 vector with *fcs* gene and above mutants to verify the enzyme activity for transforming ferulic acid to vanillin. Among all the mutants, F74W, R147Q and ΔC260-287 displayed the best enzyme activities. It is believed that these experimental results could provide valuable ideas and design guide for the biosynthesis of vanillin in the future.

## 2. Results

### 2.1. Sequence and Conservation Analysis of Ech

The targeted Ech sequence was selected from *Streptomyces* sp. strain V-1 (accession number: S5LPF1) in UniProt database. The multiple sequence alignment (MSA) of Ech were performed by retrieving 250 sequences which exhibit more than 60% identity with the targeted sequence from UniProt database. The sequence logo diagram (Appendix A) indicated several conserved regions, which were considered related to binding and catalysis between Ech enzyme and substrate feruloyl-CoA. To elucidate the essential residues which interacted with the enzyme, the substrate feruloyl-CoA could be categorized into two parts, i.e., the feruloyl group and the CoA group according to its structure. As shown in Figure 1, residues of Y83 and G160 (orange), the binding site for the feruloyl group, displayed the highly consistency in Ech enzymes, indicating their important role in the catalytic functions. Meanwhile, the proximity of feruloyl group close to the catalytic center should also be considered. The residues of R38, A40, A76–L80, W125, F127–G129, S151, E152 and W155 (green) binding with the CoA moiety displayed complete conservation, except E72. It could be recognized that these binding residues were crucial for the stability of the CoA group, and it identified that the CoA group was used to bind and stabilize the substrate. Within the binding pocket of Ech, there are both highly and partially conserved residues, so it is reasonable and practical to design and modify these conserved residues to increase the enzymatic catalytic activities.

### 2.2. Structural Features of Ech and Its Binding Mode with FCA

In order to further understand the structure and binding mode with the substrate feruloyl-coenzyme A (FCA), the Ech monomer 3D structure was established based on the hexamer crystal data of hydroxycinnamoyl-CoA hydratase lyase (Hchl from *Pseudomonas fluorescens*, PDB: 2VSS) (Appendix A). To keep the intact catalytic pocket, the Ech monomer was compared with the hexamer model, and generated the 3D structure of the Ech dimer (Appendix A). In this research, unless specified, the structure model of Ech was the dimer structure with Chain A and Chain B.

The electrostatic potentials were calculated using Adaptive Poisson-Boltzmann Solver (APBS) to visualize the electrostatic potential surface, which indicated that Chain A and Chain B of the Ech dimer together form a long and narrow cavity as the binding pocket (Figure 2a). The entrance of the binding pocket presented a higher positive potential distribution, with the basic amino acid residues on both sides (Chain A: K37, R38; Chain B: K274, R277), which is favorable for the binding of negatively charged CoA moiety. Meanwhile, to further understand the statistical potential distribution, the binding pockets of Chain A and Chain B were disassembled. It could be found that polar uncharged residues (Y83, S103, G129, S151 and W155) and M78 were distributed in the middle of the binding pocket of Chain A, which is fundamental for the binding of substrate FCA. Acidic amino acid residue E152 of Chain A is located in the catalytic center in the binding pocket, which was indispensable for the catalytic reaction. These results indicated that the binding pocket in Chain A constituted the major region for the whole binding pocket (Figure 2b). Compared with Chain A, the binding pocket of Chain B was composed of only several residues, such as the K274, R277 at and G279 near the entrance, and polar uncharged Y248 located innermost of the pocket, which was believed to stabilize the terminal of substrate FCA (Figure 2c).

The Ech-FCA complex was generated by molecular docking, with the binding mode of Ech and FCA shown in Figure 3. The binding energy of −8.5 kcal/mol generated by AutoDock Vina indicated a stable complex structure between Ech and FCA. Similarly, the binding sites in the binding pocket could be categorized into two moieties, i.e., the feruloyl binding moiety and the CoA binding moiety, corresponding to the feruloyl group and the CoA group mentioned above. The CoA binding moiety consists of quantities of hydrogen bonds and salt bridges, which is the starting point of catalytic reaction. Among these hydrophilic and charged residues, basic residues R38 and K274 (Chain B) could form stable salt bridge with the phosphor group of FCA, which is the structural element for the stable binding between FCA and Ech. At the same time, some polar residues such as S151, W155 and G279 (Chain B) as well as residues A76, M78 and L80 could bind with the CoA binding moiety by hydrogen bonds, further strengthening the binding extent of FCA. The feruloyl binding moiety involved in the catalytic reactions, the carbonyl group in feruloyl binding moiety could establish the hydrogen bonds with M78 and G129, and Y248 (Chain B) could also establish the hydrogen bonds with the phenolic hydroxyl group in the feruloyl binding moiety. The most important E152 is located above the FCA, which maintains the proper distance (4.44 Å) between the substrate and reactive site, to guarantee the initiation of the catalytic reaction. All these docking results are highly consistent with the sequence analysis, so it is practical to start the rational design on these docking results.

### 2.3. Rational Design for Ech

Based upon the above profiling for the catalytic mechanism, the primary candidate region for rational design focuses on the feruloyl binding moiety. Conservation analysis indicated that A130 and T132 are moderately conserved, which made these two loci prone to become smaller residues like Gly and Ser, so that a single mutant A130G and a double mutant A130G/T132S were designed, to enlarge the cavity for the binding pocket and strengthen the flexibility to increase the catalytic efficiency. When the Ech from *Streptomyces* sp. V-1 was performed homologous alignment with other genera, some other candidate residues could be chosen and substituted to optimize the Ech catalytic activity. Hence, F74 was mutated to Trp, R147 was mutated to Gln, and Q255 was mutated to Arg based on sequence alignment (Appendix A).

Subsequently, a 100 ns molecular dynamics (MD) simulation was performed towards the Ech-FCA complex in three independent replicas to identify the residues with unstable effects on the Ech-FCA complex. Due to the irregular structure of residues 1–14 at the N-terminus, residues 1–14 were removed at the beginning of the simulation to ensure the acceptable and validity of the initial model. RMSD and RMSF during MD simulation were shown in Figure 4a,b. It could be read from the first 50 ns stage, that the RMSD indicated a relatively stable condition, while in the next 50 ns the average RMSD was 5.64 Å which indicated some unstable factors in the wild-type (WT) of Ech. The RMSF result also verified this proposition. The RMSF indicated the most significant fluctuation region located in the 232–287 from N terminal (Chain A), which sited at the junctions of the polymer. However, residues 519–574 (232–287 in Chain B) indicated relatively smaller fluctuations. It could be speculated that the residues 232–287 in Chain B established tight interactions with Chain A and substrate FCA. Also, a relatively large RMSF was found near T90. Hence, two deletion mutants (ΔT90 and ΔTGPEIL) were designed. ΔT90 means that the T90 residue was deleted, and ΔTGPEIL means that the residues 91–95 were replaced by GPEIL besides the T90 deletion. Additionally, to decrease the influence of residues from both N and C terminals, two truncated mutants were designed as ΔN1-11 and ΔC260-287. All these designed mutants and their descriptions are listed in Table 1, and their position in protein can be found in Figure 4c.

### 2.4. Molecular Dynamics Simulation and Binding Free Energy Analysis for Ech and Its Mutants

To further understand the dynamic influence of different mutations on Ech, as well as the structural differences between mutants and wild-type of Ech, 100 ns molecular dynamics (MD) simulations were performed on all mutants. To ensure the credibility of the simulation experiments, the parametric conditions of the MD were consistent with the WT as described above. All mutants were constructed based on the initial model. The molecular mechanisms of each mutant would be reassessed through the analysis of molecular dynamics simulations. The RMSD of three independent MD simulations confirmed that all systems had reached equilibrium during the 100 ns MD simulations (Appendix A). The RMSD histograms showed the changes and distribution of the conformations of the system, and the WT appeared to have different conformational distributions (Appendix A), suggesting that the conformation of WT was unstable, thus affecting the catalytic activity. Compared to the WT, most mutations (such as F74W and Q255R) reduce the conformational distribution of the structure to some extent, indicating that appropriate mutations can stabilize the overall structure of Ech. The specific structural features and locations of the mutants are shown in Figure 5 and Appendix A. To explore the effect of mutations on catalytic activity in more detail, the MD simulation results together with the experimental validation will be discussed in the Section 3.

MM/GBSA were performed to calculate the system binding free energy for wild-type Ech and mutants. It could be read from Table 2 that the wild-type Ech exhibited a binding free energy of −102.34 kcal/mol, indicating a high binding between WT and FCA. Mutants A130G, A130G/T132S and Q255R displayed lower energies compared with WT, indicating their higher combination with the substrate FCA. While the binding free energies of F74W, ΔT90 and ΔTGPEIL were slightly reduced, and R147Q remained almost constant. Furthermore, compared to WT, mutants F74W and R147Q showed a decrease in electrostatic interaction energies (ΔE_EL_) of 75.34 kcal/mol and 35.27 kcal/mol, respectively, while polar solvation energies (ΔE_GB_) increased by 78.54 kcal/mol and 24.11 kcal/mol, respectively. On the other hand, the other mutants (except ΔTGPEIL) almost always showed large energy changes (exceeding 100 kcal/mol) in both the electrostatic interaction energy and the polar solvation energy, suggesting that the mutations significantly altered the interaction mode between Ech and FCA. The binding affinity of the truncation ΔC260-287 to FCA was drastically reduced, suggesting that the partially truncated amino acid residues at the C-terminal end play a very crucial role in the binding of the FCA, and that the significant reduction in the polar solvation energy is indicative of an increase in the surface area of the substrate exposed to the solvent environment.

Then, it is necessary to perform the per-residue energy decompositions on the total binding free energy of the system into the energy contribution of each residue, so as to elucidate/identify the residues that significantly affect the system binding free energy (Appendix A), and providing valuable data to support further discussion of the effect of each residue. All residues that exhibited energy contribution are listed in Appendix A.

### 2.5. Experimental Verification of Wild-Type Ech and Its Mutants

Wild-type Ech and all mutants were constructed and introduced into the pET-Duet-1 expression vectors, and the ferulic acid was supplemented as substrate. After 24 h of biotransformation, the resulting vanillin was tested by GC and the productivity was plotted in Figure 6.

As shown in Figure 6, the single mutants F74W and R147Q with residue substitutions both showed an approximately 2-fold increase in activity over WT, whereas the Q255R mutant lost all catalytic activity. In addition, the N-terminal truncation ΔN1-11 showed an increase in catalytic activity, suggesting that the disordered motif at the N-terminus has an effect on the catalytic activity of the enzyme. In contrast, the C-terminal truncation (ΔC260-287) showed the opposite effect, indicating that the residues in the C-terminal were necessary for the catalytic activity of Ech. Unfortunately, neither the A130G nor the A130G/T132S mutants near the active site showed higher catalytic activity, which may indicate a potentially critical role of A130 for catalytic activity. Lastly, the two deletion mutants ΔT90 and ΔTGPEIL also show reduced catalytic activity, indicating that a deletion mutation at site 90 is not a favorable option.

Among these mutants, there are only three positive mutants displayed better productivitity, one with decreased performance, and other with no products generated. The detailed analysis will be discussed subsequently.

## 3. Discussion

Directed evolution is classical method for modifying the target enzyme activities for years [26,27,28]. Rational designs are strongly dependent on the enzyme structure and bioinformatics methods. In recent years, the molecular mechanisms of a large number of enzymes have been elucidated by molecular dynamics simulation methods [29,30,31], thus enabling the rational design of enzymes. In this study, Ech, a key enzyme in the biosynthesis pathway of vanillin, was selected to improve its catalytic activity through rational design. The sequence and structure features of Ech and its binding mode to FCA were investigated by bioinformatics techniques such as sequence comparison, molecular docking and molecular dynamics simulation. Nine mutants (F74W, A130G, A130G/T132S, R147Q, Q255R, ΔT90, ΔTGPEIL, ΔN1-11 and ΔC260-287) were rationally designed based on sequence and structure guidance, of which three mutants (F74W, R147Q and ΔN1-11) were eventually confirmed to have enhanced activity through experimental validation. Here, all mutants were divided into four groups according to the type of design, and the main factors affecting and improving the catalytic activity of Ech are discussed by comparing the computational data with the experimental results.

For the group with residues substituted from *Pseudomonas fluorescens* and *Amycolatopsis thermoflava*, F74W and R147Q exhibited positive vanillin production, indicating the higher Ech activity, whereas the other mutant Q255R from *Pseudomonas fluorescens* was completely inactive. Among them, both mutants F74W and R147Q displayed more than twice Ech activity than WT. Despite F74 and R147 being located some distance away from the active pocket, and not interacting with the substrate FCA directly, there are some indescribable effects which may increase the Ech activity. Also, F74W could interact with many more residues to establish the hydrogen bond and hydrophobic interactions, including A130, L32, L67, V122 and L134 (Figure 5b). Comparably, the F74 residue in the WT only interact with A120 and V122 with hydrophobic interaction. However, it seems that R147Q did not increase the Ech activity by strengthening the autologous interactions. The simulation results indicated that due to the elimination of autologous charge, the intrinsic salt bridge which connected the guanidyl of R147 to the carboxyl of charged D187 could be changed to connect to another basic R189, with hydrogen bond generated in the residue itself (Figure 5a). These changes seemed beneficial for the Ech catalytic activity towards FCA. Meanwhile, F74W displayed a lower binding free energy (−94.07 kcal/mol) than the WT, demonstrating the reduction of electrostatic interaction energy and increase of polar solvation energy. Despite the fact that R147Q displayed some electrostatic interaction energy reduction, the total binding energy (−102.99 kcal/mol) is similar to the WT. However, the residue Gln, which substitutes the Arg, demonstrated a smaller side chain, which may cause the substrate some directional and/or positional changes so as to increase the Ech catalytic activity. Compared with the WT, V162/V163 and G161/V162 energy contributions were identified in Chain A of F74W and R147Q (Appendix A). As for the inactive mutant Q255R, the simulated conformation showed that Q255R formed a new salt bridge interaction with residue D88 on the other α-helix (Figure 5d), which appeared to promote structural stabilization. Further investigation revealed, however, that this interaction was not beneficial for the catalytic reaction. Since Q255 is close to the release channel of the product, an overly rigid structure may result in a difficult release of the product, leading to a decrease in catalytic activity.

For the group that modifyed the size and flexibility of the active cavity, mutants A130G and A130G/T132S generated little vanillin production. This unexpected result suggested that the A130 was essential for the catalytic activity, even a minor modification could inactivate this role. This unfavorable result may be partially due to these two residues being too close to the catalytic center (Figure 5c). Also, A130G demonstrated similar binding free energy distribution with A130G/T132S. Compared with WT, both mutants displayed significantly lower binding free energy and higher polar solvation energy, indicating the prominent changes in the interaction between the FCA and Ech, so as to deactivate the enzyme. Also, A130G lost the A76 interaction in Chain A, and L567 interaction in Chain B. Despite the W125/F127 provided better energy contribution in A130G/T132S, only the R564 in Chain B kept original energy (Appendix A). These changes finally rendered the deactivation of Ech.

For the group with reducing structure flexibility, ΔT90 exhibited lower vanillin production, indicating lower Ech activity. Both the deletion/multiple mutants ΔT90 and ΔTGPEIL exhibited the configuration changes for the vanillin release tunnel. The B-factor color (Figure 5e) indicated more significant flexibility changes for the ΔT90, while the ΔTGPEIL greatly decreased the flexibility. This is crucial so that ΔT90 still demonstrated some Ech activity but ΔTGPEIL did not. The binding free energy of ΔT90 and ΔTGPEIL is similar to WT, with lower binding strength. Partially due to the conformation changes rendered by the mutation, the substrate position in the Ech could be changed so as to hamper the catalytic activity. Compared with the WT, ΔT90 and ΔTGPEIL almost kept all the essential residues, and it could be calculated that the W125 and F127 in Chain A exhibited more energy contribution (Appendix A). So, the activity of Ech decrease and loss may be attributed to the regional structure change.

For the group with N and C terminal truncation, ΔN1-11 also exhibited positive vanillin production, indicating higher Ech activity. This result indicated that the N terminal unordered motif is important for the entire enzyme stability, the elimination of these unorder motifs could greatly increase the Ech stability. On the contrary, the C terminal truncation ΔC260-287 could not catalyze the vanillin production, indicating that this C terminal is indispensable for the enzyme activity. As shown in Appendix A, the loss of those truncated residues renders the significant conformation changes, especially in the CoA binding moiety, which made the substrate FCA unable to keep a stable configuration during the catalytic reaction process, so as to deactivate the enzyme. Meanwhile, for the ΔC260-287, it could be found from the energy decomposition that all the residues that could provide energy contribution were truncated in Chain B (Appendix A), so the binding between Ech and FCA were quite unstable, that the catalytic activity were greatly hampered.

In summary, the residues that affected the binding free energy between Ech and FCA could be mainly categorized into two regions, such as K37, R38, A76, M78, L80, W125 and F127 in Chain A, and F557, F561 and R564 in Chain B. Among these residues, the basic residues K37, R38, K561 and R564 demonstrated higher energy contribution in most systems, which indicated their important roles in the binding of Ech and FCA. According to the previous simulation, A76, M78, and L80 could form a stable binding region, which could impose the FCA in the binding pocket. Additionally, non-polar residues W125 and F127 could contribute to the hydrophobic interaction with FCA, so as to strengthen the binding of FCA.

In this study, our rational design strategy is reasonable and verified by the experiment results. Three mutants, F74W, R147Q, and ΔN1-11, displayed a positive increase inthe production of vanillin, which provided the basis for rational engineering of the enzyme to improve the vanillin biosynthesis. The MD simulation results could partially explain the unfavorable results with mutation location and stability by hydrogen bonds, hydrophobic interactions etc. However, there are still some unresolved issues, e.g., for the mutants A130G and A130G/T132S, it is still unclear which specific effects of residues A130 and T132 on the catalytic activity of the enzyme. Although there is evidence to suggest that the neighboring residue G129, as an important residue in the formation of the oxyanion hole (OAH) in active sites, may be affected by mutations in neighboring residues, how small changes affect the catalytic activity of Ech could be probably unobservable with classical MD simulation methods. Perhaps quantum mechanics (QM) or quantum mechanics/molecular mechanics (QM/MM) methods could be used in future studies to further investigate the effect of mutations on the catalytic mechanism of Ech.

From this analysis, it could be concluded that the effect of enzyme stability is a crucial factor to be considered when improving the catalytic activity of enzyme catalysis through rational design. Therefore, it is necessary to balance the catalytic activity with the enzyme stability. This rational design strategy could also enlighten other studies for enzyme optimization in the future, and further work was desired to accomplish better results.

## 4. Materials and Methods

### 4.1. Bioinformatics Analysis

#### 4.1.1. Multiple Sequence Alignment and Conservation Analysis

Two key enzymes of the CoA-dependent pathway of vanillin biosynthesis from ferulic acid from *Streptomyces* sp. V-1 were collected from the UniProt database. One is the ferulic acid synthase (Fcs, accession number: S5M744), and the other is the enoyl-CoA hydratase/lyase (Ech, accession number: S5LPF1). Sequence alignments were performed towards Ech, sequences with more than 60 percent homology were retrieved, and the ClusterW [32] algorithm was used to calculate the multiple sequence alignment. The Jalview2.11.2.6 [33] software was applied for the homologous alignments; the sequence log was generated by the online website WebLogo3 [34] (https://weblogo.threeplusone.com (accessed on 14 March 2023)).

#### 4.1.2. Homologous Modeling and Molecular Docking

By retrieving Ech sequence (UniProt: S5LPF1) from Uniprot, Blast results indicated the optimum template for Ech modeling was hydroxycinnamoyl-CoA hydratase lyase (Hchl, PDB ID: 2VSS). Modeller9.23 [35] was utilized to construct the homologous model. By listing the resultant Discrete Optimized Protein Energy (DOPE) value and Molpdf rate, a final 3D model was screened and determined. As the entire active pockets were involved in two subunits, an Ech dimer model was then constructed. The Ech and substrate FCA complex was constructed by AutoDock Vina 1.1.2 [36]. The docking box was set as 23 × 14 × 8 with 0.375 grid spacing. All data were calculated three times to generate the best results.

#### 4.1.3. Molecular Dynamics (MD) Simulation and Binding Free Energy Calculations

All the Molecular dynamics (MD) simulations of the enzyme-substrate complex were performed by AMBER20 [37] package. The force field parameter was set as AMBER-ff14SB [38] for protein, and the general AMBER force field (GAFF) [39] was applied to FCA. The bond angle parameters and resp charge of the small molecule were calculated by Gaussian09 [40]. The topology and coordinate files of the enzyme-substrate complex were obtained by the Tleap module from AMBER20. The whole system was dissolved in a cubic TIP3P water box with periodic boundaries, and Na^+^ and Cl^−^ were used to maintain the electric neutrality of the system. Gradient minimization is employed in the initial system to perform energy minimization so as to eliminate unfavorable contacts and structures. Then, the system would be heated slowly in NVT system by from 0 K to 300 K by a 10 kcal/(mol·Å^2^) constraint force, and balanced in NTP system for 300 ps, to keep the temperature and pressure stabilized at 300 K and 1 atm (standard atmosphere). Finally, three independent 100 ns production MD simulations (total of 300 ns) was performed in a well-balanced system with a 2.0 fs time step, and all enzyme-substrate complex systems were calculated under the same parameters and conditions. The particle-mesh Ewald (PME) [41] algorithm were used for system energy minimization and MD simulation process with a cutoff value set at 10 Å, so as to calculate the long-range electrostatic interactions throughout the whole simulation. The SHAKE [42] algorithm was applied to constrain all covalent bonds with hydrogen and to control the temperature change during the MD simulation under Langevin dynamics [43] with a collision frequency γ of 1.0.

The molecular mechanics/generalized Born surface area (MM/GBSA) method [44] was applied in analyzing the molecular interaction between enzyme and substrate. The Python script MMPBSA.py in AMBER20 was used for the calculation of the binding free energies and the per-residue energy decomposition of Ech and FCA. A total of 2500 snapshots were extracted from the last 50 ns of the MD simulation to calculate the binding free energy of MM/GBSA. The most representative conformation for each complex was selected using a clustering algorithm [45]. In addition, an energy decomposition of each residue was performed to evaluate the energy contribution of each residue in the system.

### 4.2. Experimental Verification

#### 4.2.1. Strains, Plasmids and Reagents

Synthetic *fcs* and *ech* gene sequences were optimized, manufactured and cloned into *E. coli* BL21(DE3) with *BamHI*/*HindIII* and *BglII*/*XhoI* (TaKaRa Bio Inc., Dalian, China). The recombinant stains were used for expression and whole-cell biocatalysts. The rationally designed mutant gene sequences were amplified by PCR with the original *ech* primer pairs (Table 3). All Ech mutants were verified by sequencing services provided from Genscript (Nanjing, China). All reagents used herein are analytical and chromatographic grade.

#### 4.2.2. Whole Cell Catalysis

40 µL of WT and mutant strains were inoculated into 4 mL LB media supplemented with 50 ug/mL Amp resistance. Cells were cultured at 37 °C overnight with 220 rpm. Then 1% inoculation was transferred into a fresh LB media with the same resistance, and cultured for 4–6 h with the same condition until the OD_600_ accumulated to 0.6~0.8. Next, the final concentration of 0.2 mM of IPTG was supplemented, and cells were cultured at 16 °C for 16 h to induce the enzyme expression. After the induction stage, cells were harvested by centrifugation at 3500 rpm for 10 min, and washed twice with 10 mL sterilized PBS solutions. Finally, 10 mL cells were transferred into the 50 mL conical flask and final concentration of 5 mM ferulic acid was supplemented, and cultured at 30 °C, 200 rpm for 24 h.

#### 4.2.3. Product Identification and Analysis

The reactions were stopped by adding 20 μL of 6 M HCl, and an equal volume of ethyl acetate was fully mixed for 1 min to perform the extraction. Then, the mixtures were centrifugated at 1200 rpm for 10 min. The supernatant was used for the GC analysis. GC analysis was performed on Varian 456 Gas chromatograph equipped with HP-5 capillary column (30 m × 0.320 mm × 0.25 μm). The detection conditions were set 2 min at 140 °C, then increased to 220 °C at 15 °C/min and kept at 220 °C for 2 min. The injector temperature was set at 250 °C, the FID detector temperature was set at 280 °C, and the injection volume was 5 µL.

## 5. Conclusions

In this study, Ech was rationally designed based on sequence and structural features, and a series of mutants, including F74W, A130G, A130G/T132S, R147Q, Q255R, ΔT90, ΔN1-11, ΔC260-287, and ΔTGPEIL, were obtained, among which the mutants F74W, R147Q, and ΔN1-11 showed an approximately 2-fold increase in activity compared with the WT. The MD simulation results indicate that the basic residues K37, R38, K561, and R564 play critical roles in the binding of Ech and FCA. Therefore maintaining the stability of these residues is essential for the catalytic activity of the enzyme. In addition, the significant changes in the electrostatic interaction energy and polar solvation energy caused by the mutations could further reduce the enzyme activity. Our study provides valuable theoretical support and a reliable basis for the rational design of enzymes and the industrialization of biosynthesized vanillin. In the future, we will further improve and explore the molecular mechanisms affecting the catalytic activity of Ech to enhance the catalytic activity of Ech and thus promote the development of the vanillin industry.

## Data Availability

The data presented in this study are available on reasonable request from the corresponding author.

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
