# Peer review of "Biosynthesis of Vanillin by Rational Design of Enoyl-CoA Hydratase/Lyase"

_ijms, 2023, doi:10.3390/ijms241713631_

Round 1

Reviewer 1 Report

The current study encompasses both experimental and computational investigations. A central aspect of concern pertains to the authors' reliance on single trajectory simulations. It is important to underscore the stochastic nature inherent to simulations. Basing the assertion of the Ech wildtype enzyme's instability solely on individual trajectory simulations raises significant reservations. To establish a more robust foundation for their claim, it is recommended that the authors adopt a minimum of three simulations. This approach would better capture the probabilistic nuances underlying enzymatic changes and mitigate the potential for drawing overly assertive conclusions without adequately comprehensive evidence.

Author Response

Dear editor and reviewer 1:

Thank you very much for your evaluation and comments of our paper. Please find our reply below. Your reviewing and criticising are highly appreciated.

Please see the attachment for specific details.

Reviewer 2 Report

This research investigates the vanillin CoA-dependent (Fcs and Ech) pathways. Fcs and Ech from Streptomyces sp. strain V-1 are studied via computational methods, resulting in strategic mutations that yield enhanced vanillin production. The author used Molecular dynamics simulations to identify the key amino acids influencing Ech-FCA binding energy. The study contributes vital insights and experimental data to the vanillin biosynthetic pathway. However, there is some minor modification and clarification that will be addressed before its final publication. Please revise the manuscript according to the following comments.

General Comments: 

Abstract: Abstract was concise and well written. 

Keywords: Keywords looks fine 

Introduction: 

Line 38: change Highly to High or higher

Line 49: remove the word “one”

Line 63: The author needs to write more about the rationale/hypothesis of this research 

Line 66: replace the word variety with various

Line 75: it is the accession number, not access number

Line 83 and Figure 1: The Y248 is not conserved in all of the sequences. the Delftia acidovorans has valine (V) instead of Tyrosin (Y). The author should mention this.

Materials and Methods: 

4.1: It would be better to change the headings to Bioinformatics analysis instead of the Dry Lab part.

Figures

All of the Figure legends should be elaborate, independent, and include all the information. include the statistical tools used. The figure should be stand-alone.

Technical comments: 

1. Author mentioned that both Ech and FCA have only performed the rationale design for Ech. What was the rationale behind this?

Language Usage: Kindly revise the manuscript carefully to eliminate some minor grammatical errors. Use scientific language and try to avoid colloquial language. Use abbreviations and symbols (degree Celsius, mm, h. min, etc.) uniformly throughout the manuscript.

Kindly revise the manuscript carefully to eliminate some minor grammatical errors. Use scientific language and try to avoid colloquial language.

Author Response

Dear editor and reviewer 2:

Thank you for your comments and evaluation of our paper, we are glad to have your valuable suggestions. We have carefully read and studied your comments, and made corresponding changes to this manuscript.

Please see the attachment for specific details.

Reviewer 3 Report

d. Mechanism Conclusions: Explain what new information you have gained on the mechanism of vanillin biosynthesis by designing the enoyl-CoA hydratase/lyase. Has your research helped to better understand this process?

e. Relevance of Results: Discuss the importance of your results in the context of the field of science in which you work. Has your research influenced the development of knowledge in a given field?

f. Practical Implications: If your research has potential practical applications, highlight those implications. This may be in the production of vanillin or other areas.

g. Study Limitations: Openly discuss any limitations of your study, such as methods that may not be ideal or other difficulties that may have affected the results.

h. Future Research Directions: Suggest possible future research directions that may arise from your work. What questions remain unanswered?

i. Summary of Conclusions: At the end of the conclusion section, summarize the most important points that have been discussed in this part of the work.

j. Conclusions: In this final part of the conclusions, you can express the general conclusion of your work and its relevance to your study or practice.

k. Conclusions should be clear, logically organized and focused on the most important points. They should also emphasize the importance of your work and indicate what innovations it has brought to the field of science

Minor English language edition required.

Author Response

Dear editor and reviewer 3:

We are very grateful for your careful reviewing and comments. There valuable suggestions could significantly help us improve our manuscript. Your organized instructions are highly appreciated. After studying your detailed suggestions carefully, we have made corresponding correction and explain for the unclear description.

Please see the attachment for specific details.

Round 2

Reviewer 1 Report

The authors have made changes suggested by me, therefore, I recommend acceptance of this manuscript in its present form.

Reviewer 3 Report

Dear Authors,

The thesis was corrected according to the reviewer's suggestions and indications. The authors meticulously approached the review, refined the work and improved its quality.

Minor editing of the English language is required.